# A density-based enrichment measure for assessing colocalization in single-molecule localization microscopy data

Aske L. Ejdrup [1] ✉, Matthew D. Lycas[1], Niels Lorenzen[1], Ainoa Konomi[1], Freja Herborg [1], Kenneth L. Madsen[1] & Ulrik Gether [1] ✉

Dual-color single-molecule localization microscopy (SMLM) provides unprecedented possibilities for detailed studies of colocalization of different molecular species in a cell. However, the informational richness of the data is not fully exploited by current analysis tools that often reduce colocalization to a single value. Here, we describe a tool specifically designed for determination of co-localization in both 2D and 3D from SMLM data. The approach uses a function that describes the relative enrichment of one molecular species on the density distribution of a reference species. The function reframes the question of colocalization by providing a density-context relevant to multiple biological questions. Moreover, the function visualize enrichment (i.e. colocalization) directly in the images for easy interpretation. We demonstrate the approach's functionality on both simulated data and cultured neurons, and compare it to current alternative measures. The method is available in a Python function for easy and parameter-free implementation.

Fluorescence light microscopy (LM), combined with immuno-cyto- or -histochemistry, has been used extensively for studies of colocalization of different cellular components. These experiments have been central to our understanding of the cell, where the function of a molecular species is partially inferred from spatial association to a reference protein or organelle. Accordingly, a host of methods for quantitative colocalization analysis for wide-field image application have been developed[1]. Common to these is the conceptual reduction of colocalization to a single value. This has been a rational approach, given that the ~250 nm diffraction limit of light is orders of magnitude larger than individual molecules[2]. With the advance of novel single-molecule localization microscopy (SMLM) techniques, molecular positions can be resolved down to <20 nm by temporally separating spatially overlapping light sources[3–5]. With the resolution approaching molecular sizes scientists have been provided with a tool to study nanoarchitecture and carry out more detailed colocalization studies.

But no new venture is without new woes. SMLM techniques produce coordinate-based molecular positions, rather than the intensity-based pixel information of regular LM. This change in data type makes conventional analyses difficult to apply. Alongside, non-uniform protein nanoscale distribution in cellular membranes and cytoplasm is becoming increasingly apparent[6,7], which makes it relevant to assess colocalization in the context of protein density. Lastly, as molecular localization precision is approaching that of individual molecule size, and certainly within the size of most molecular labels, a simplistic view on colocalization is arguably a remnant from the diffraction-limit-derived conceptual reduction of the phenomenon. By expanding the colocalization analysis to include biologically relevant context, the data might further address emerging scientific questions.

While several methods have been developed to assess colocalization in super-resolved microscopy data, they either reduce the measure to a single value, which may limit biological insight, or depend on user-defined input parameters. Rossy and colleagues[8], coordinate-based colocalization (CBC)[9] and cluster detection with the degree of colocalization (ClusDoC)[10] all employ parameter-based approaches where user-defined values strongly influence the outcome[11], and neither ClusDoC nor Rossy and colleagues approach work with 3D data. While CBC does include a density context in its calculations, the

[1]Department of Neuroscience, Faculty of Health and Medical Sciences, University of Copenhagen, Copenhagen, Denmark. ✉e-mail: ejdrup@sund.ku.dk; gether@sund.ku.dk

resulting value incorporates the local density of both species and compresses the information to a summary statistic between −1 and 1 that provides rather little information on where the interaction takes place. Coloc-Tesseler[12] works in 3D and is density based, but likewise it outputs a summary statistic that reduces colocalization to a single value without further context to the value.

In this work, we present a function that describes relative enrichment of one molecular species on the density distribution of a reference species. The algorithm is inspired by the pair-correlation function[13] and the Voronoï tessellation of Coloc-Tesseler[12]. It reframes the question of colocalization by providing a density-context relevant to numerous biological questions. The tool is parameter-free and outputs a unitless ratio measure based on the average distance to neighboring proteins. Further, the algorithm can visualize enrichment directly in the images for qualitative interpretation, making colocalization information from super-resolution microscopy interpretable by scientists without advanced microscopy expertise. It works in both 2D and 3D and is available as a python function for further open development and easy implementation in a processing pipeline across operating systems and file formats. We envision this tool broadly used in future studies of the cellular nano-architectural organization due to its intuitive, unitless representation of colocalization, its contextual insight when protein distribution is heterogenous, its ability to

generate informative qualitative images, and its easy and parameter-free implementation.

## Results

### Tessellation-based assessment of relative enrichment

Voronoï tessellation partitions a plane into discrete regions around seeds with each region containing the area closest to its seed (Fig. 1). The method can be applied to SMLM data, where each molecular localization represents a seed, and has proven itself a robust and efficient tool to assess local protein density[14–16]. Previously, Levet and colleagues proposed a colocalization analysis based on tessellation[12]. Building on their approach, we propose a method that integrates tessellation and conceptual elements of the pair-correlation function[13], with the aim to unfold new biological insight by deriving a continuous measure of colocalization in the context of density. Our method calculates the enrichment of one primary molecular species across the density distribution of a reference molecular species. The result is a histogram of densities for the reference species with a score for each bin, indicating how enriched the primary species is near reference species with the corresponding density. The two species can be interchanged depending on the biology of interest.

To illustrate the method, we simulated two randomly distributed species of localizations (Fig. 1a). First step is to split the assigned reference species into separate regions by Voronoï tessellation (Fig. 1b). We refer to these as reference regions. Regions with no outer neighbor are considered edge regions, denoted with dashed lines and are not included in the analysis. Localizations of the primary species are then superimposed in their original positions (Fig. 1c), and the heart of the method is to quantify the observed number of primary localizations for each reference region, and compare it to the expected amount:

$$\text{relative enrichment} = \frac{\text{primary}^{\text{observed}}}{\text{primary}^{\text{expected}}} \qquad (1)$$

The expected number of primary localizations is calculated for each region as per Eq. (2), by dividing the individual region area with the total area of all regions, and then multiplying that number with the total number of observed primary localization:

$$\text{primary}^{\text{expected}} = \frac{\text{area}^{\text{region}}}{\text{area}^{\text{total}}} \, \text{primary}^{\text{total}} \qquad (2)$$

To exemplify this, the relative enrichment (RE) is calculated for the region denoted with an asterisk on Fig. 1c (Fig. 1e). This region has an area of 0.24 μm², with the total area of all nine included regions at 1.3 μm². That is 18.5% of the total area, and with 20 total primary localizations, Eq. 2 tells us that the expected occurrence is 3.7 localizations. As only one primary localization is found inside that region, Eq. (1) gives us a RE score of 0.27. For two completely uniform distributions, each region of the reference species would have an average RE of 1, and as such an RE below 1 indicates a depletion. Conversely, an RE above 1 indicates an enrichment of the primary species in the vicinity of the associated reference localization. When plotting all the regions of the reference species color-coded by their RE, the heterogenous distribution of the primary species relative to the reference species becomes evident (Fig. 1d).

### Larger dataset and scoring across densities

A typical SMLM image contains on the order of thousands to millions of localizations. To distil information from a wealth of RE scores in a density context, our method bins reference localizations by their region size, and assess the mean enrichment score per bin. To demonstrate this, we simulated normally-distributed, overlapping clusters of each species, slightly off-set from one another, and added

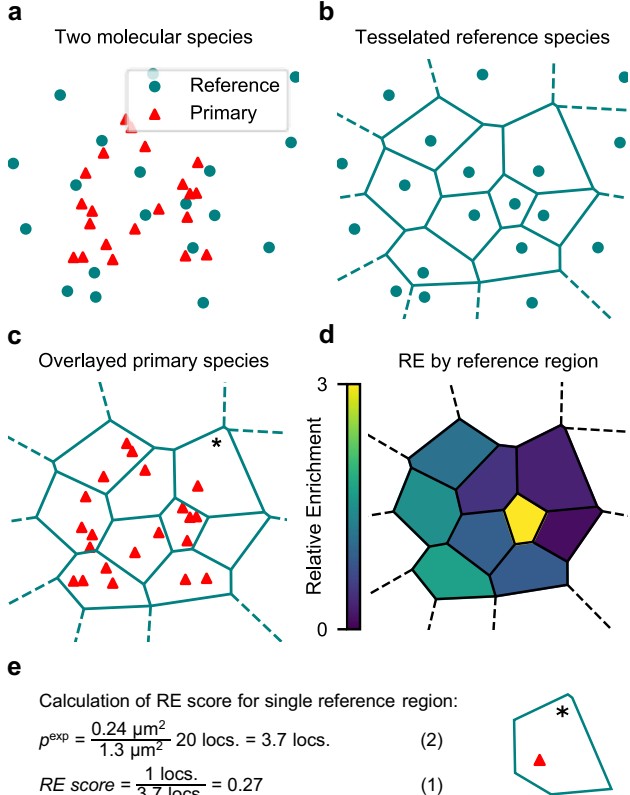

**Fig. 1 | Tessellation-based assessment of relative enrichment. a** Simulated random distribution of two molecular species. For computation of relative enrichment (RE) one is assigned as the primary species and one as the reference species. RE can be bidirectionally computed and will by design yield different results if the two species are switched. **b** Voronoï-regions from tessellation of the reference species from (**a**). **c** Primary species superimposed on the Voronoï-regions of the reference species. **d** Voronoï-regions for the reference species, color-coded by their individual RE scores. All calculated as in panel (**e**). **e** Calculation of single RE value from Voronoï-region marked with (\*) in c, based on the distribution of the primary species.

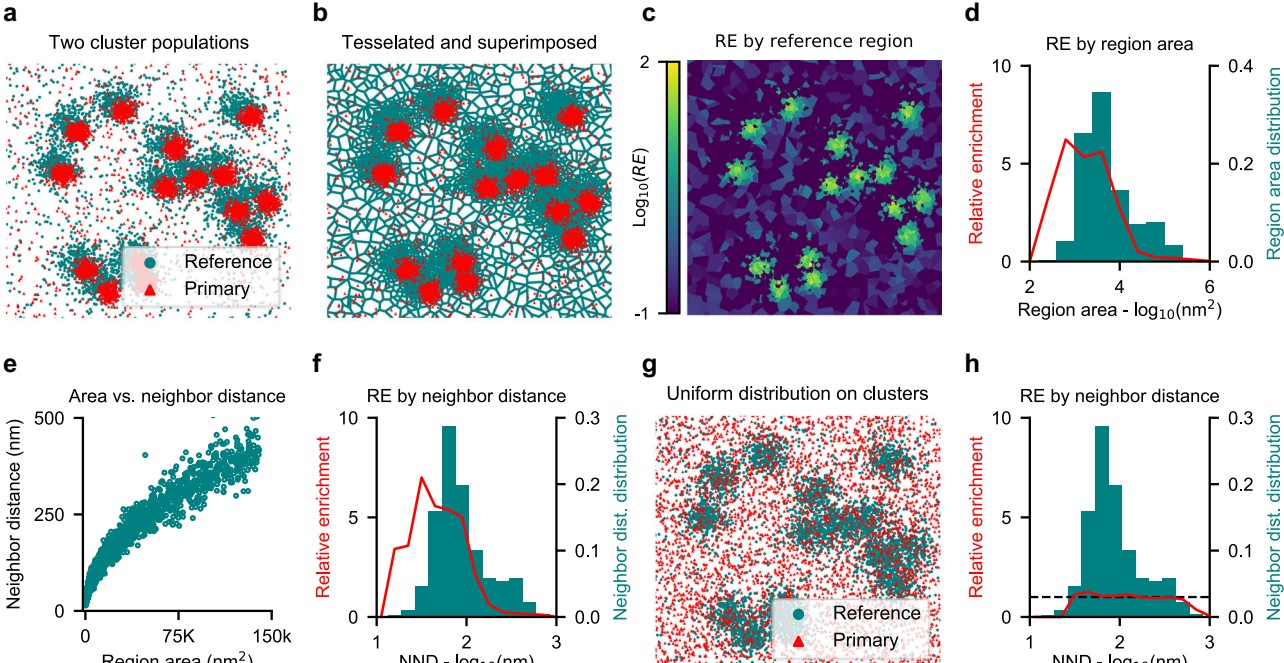

**Fig. 2 | Visualising RE across densities for larger images. a** Simulation of two colocalizing populations of normally distributed clusters with random background noise. One species of clusters is slightly offset from the other. **b** Voronoï-regions of the reference species from (**a**), with the primary species as overlay. **c** Voronoï regions for the reference species, color-coded by their individual RE scores, calculated based on the distribution of the primary species. **d** Mean RE score across reference densities. Reference regions are binned by area, and mean RE score calculated for each bin. RE score plotted on left-hand y-axis, and relative bin distribution plotted on right-hand y-axis. **e** Comparison of region area versus mean distance to nearest neighbours for the reference species. **f** Mean RE score across reference densities, binned by nearest neighbour distance (NND), rather than area. **g** Simulation of two protein populations: a clustered and a uniform distribution. **h** Same plot as (**f**), but for the uniform primary species distribution data shown on (**g**). Dashed line indicates an RE of 1.

random background noise (Fig. 2a). The RE score for each region was assessed as described earlier (Fig. 2b), and an illustrative RE-color-coded map of reference regions is shown on Fig. 2c. Following this procedure, we uniformly binned regions by the logarithm of their area. This generates a histogram that visualize the density distribution of the reference species (Fig. 2d). For each bin, we additionally plot the mean RE of the primary species in reference regions of that size interval. This results in a plot that shows the relative enrichment of one molecular species for a given density of a reference molecular species. This presentation of RE by reference region size provides a density context to colocalization, without reducing it to a one-dimensional phenomenon.

Binning regions by area is the most apparent choice, as RE is computed from area. However, area is not an immediately intuitive measure[17]. Therefore, we introduce binning by distance to neighbors, which is a more intuitive parameter, directly correlated to the size of the tessellated area. As complex polygons have no direct radius, mean distance to nearest neighbors is an excellent alternative, that provides a one-dimensional measure for local density. Fortunately, nearest-neighbor distance (NND) scales rather robustly with region area (Fig. 2e), with the added advantage of higher resolution in the densest localizations, due to the concave shape of the correlation. As shown on Fig. 2f, binning by NND rather than area produce a similar result, but with a more intuitive unit on the x-axis.

To show an example with no colocalization, we once again simulated two species with different distributions: one clustered and one uniform (Fig. 2g). The irregular distribution in one species does not distort the result, and this situation still yields an average RE score of 1 across densities (Fig. 2h), as the method accounts for the area covered when computing enrichment. This is true regardless of which species is chosen as reference (Fig. S1a). While some of the smaller regions might score high in RE, this is counteracted by a low score for the majority, and the two will even out in the binning when no specific colocalization is present. This is illustrated by the color-coded plot for RE score by region (Fig. S1b). The bimodal distribution of the reference species is also apparent from the histogram of NND, showing the usefulness of mapping densities when assessing colocalization.

To assess the outcome of various scenarios, we simulated data and calculated the RE values. Three pairs of random distributions with increasing densities were simulated first (Fig. S2a–c), which all yielded an RE of 1 across local densities. We also simulated a set of randomly distributed as well as anti-colocalized clusters with background noise in both (Fig. S2d, e). Akin to the random distributions, random clusters resulted in an RE of 1 across localization densities, whereas anti-colocalized clusters gave a lower RE score for the high-density localization, i.e. the clusters, and an RE score above one for the non-clustered portion. Finally, we simulated two identical random distributions (perfect colocalization of each pair, Fig. 2f). Here, the RE by design yields the inverse of the mean area of the density bin, as this is an indicator of how likely a localization is within the given region and relative enrichment function does not consider the distance from seed of the reference region to the primary localization.

Coltharp and colleagues point out that careful consideration should be put into the selected region size when assessing spatial organization[18]. This can significantly impact the colocalization score of many methods, and lead to unwarranted conclusions if neglected. While region size can impact absolute RE values, the scoring across density bins in this method carries information about this spatial distribution (Fig. S1c–f), rather than incorrectly influencing a single measure of colocalization. This further highlights the usefulness of the measure.

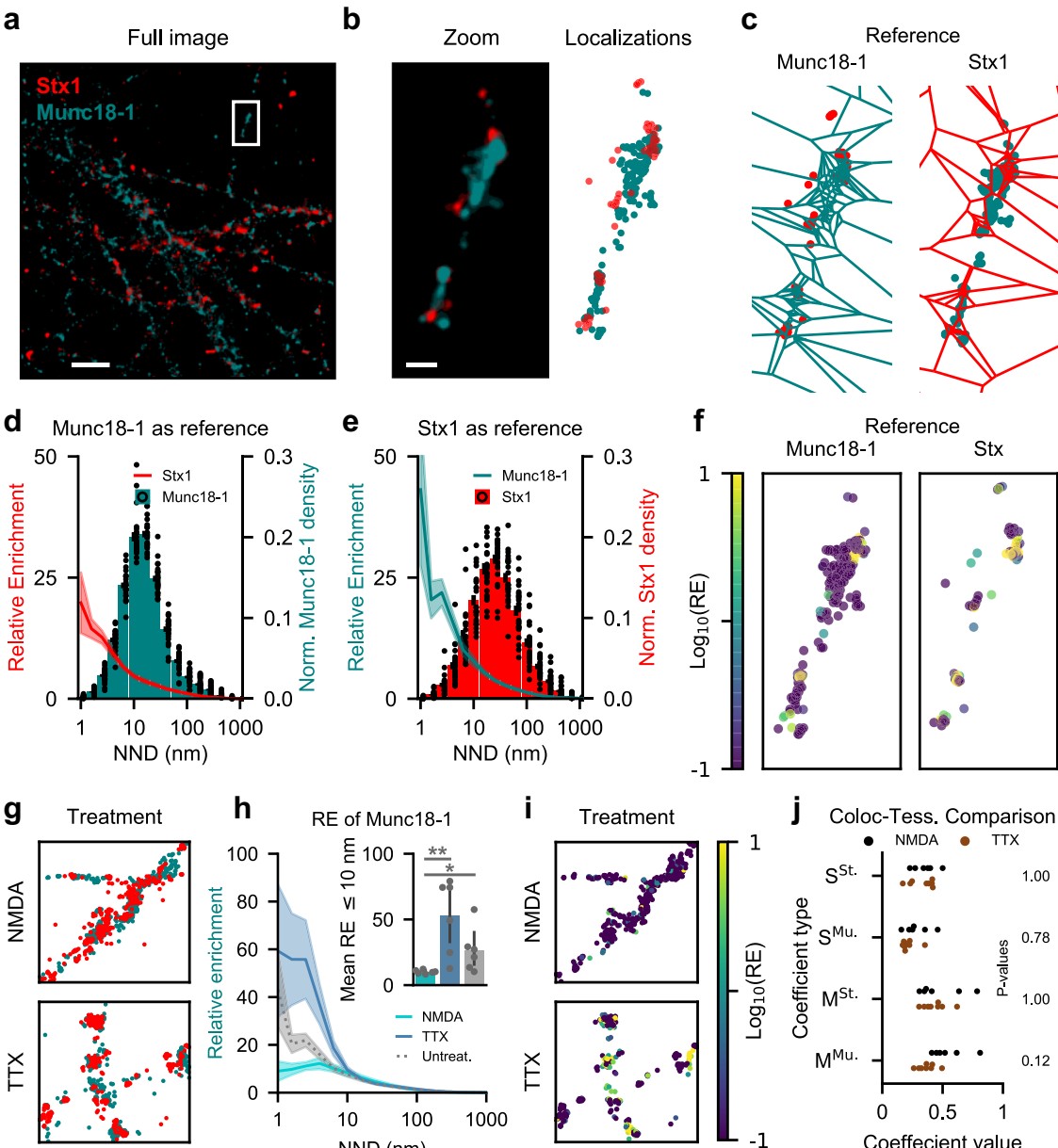

**Fig. 3 | Visualising RE across densities for larger images. a** Representative image of Stx1 and Munc18-1 in primary hippocampal neurons acquired with dSTORM. Molecular localizations visualized by gaussian representation. Scale bar is 4 μm. **b** Zoom on single varicosity from (**a**), with both Gaussian representation (left) and individual localizations (right). Scale bar is 400 nm. **c** Voronoï-regions for tessellation of either Munc18-1 (left) or Stx1 (right). Individual localizations of opposite species are plotted beneath. Same scale as (**b**). **d** Munc18-1 regions binned by nearest neighbour distance, with mean Stx1 RE value for each bin as line plot. RE score on left-hand y-axis, and relative bin distribution on right-hand y-axis. Shaded area and black bars indicate S.E.M., $n = 8$ images. **e** Same as (**d**), but with reference and primary species reversed. **f** Localizations of the reference species color-coded by RE score for Munc18-1 (left) and Stx1 (right). **g** Representative axons of

hippocampal neurons after treatment with NMDA (top) or TTX (bottom). **h** Relative enrichment of Munc18-1 across Stx1 densities after NMDA (dashed, $n = 7$ images) or TTX (dotted, $n = 8$ images) treatment. Shaded area indicates S.E.M. Inset: Mean RE of regions with an NND ≤ 10 nm for NMDA, TTX or untreated cultures (Untreat:NMDA., $p = 0.004$; Untreat:TTX, $p = 0.084$; NMDA:TTX, $p = 0.039$. Unpaired, two-tailed student's t-test by image, FWER correction with Bonferroni-Holm). **i** Images from (**g**), with Stx1 localizations color-coded by RE of Munc18-1. **j** Spearman (S) and Mander's (M) coefficient computed as per[12] (H$_0$: NMDA = TTX. S$^{Stx}$, $p = 1.00$; S$^{Munc}$, $p = 0.78$; M$^{Stx}$, $p = 1.00$; M$^{Munc}$, $p = 0.12$. Unpaired, two-tailed student's t-test by image, FWER correction with Bonferroni-Holm). Source data are provided as a Source Data file.

## Assessing colocalization of synaptic release proteins

Molecular architecture is highly important for the correct functioning of cells. Especially in neuronal synapses, where various proteins coordinate docking, priming and fusion during synaptic vesicle release at astounding precision and speed[19]. To showcase the insight supplied by applying our relative enrichment measure, we imaged two proteins central to vesicle fusion: mammalian uncoordinated 18 (Munc18-1) and syntaxin-1 (Stx1). Stx1 has multiple functions and is involved in both

neuronal maintenance as well as vesicle docking and fusion[20]. In contrast, Munc18-1 exclusively orchestrate the assembly of release machinery[21]. Evidence from previous studies in neuroendocrine and neuronal cells suggest the two proteins colocalize at docked vesicles and disperse following vesicular neurotransmitter release[22–27]. We harvested hippocampal neurons from rat pups, fixated the cultures, stained for the two proteins and imaged with direct stochastic optical reconstruction microscopy (dSTORM)[28] (see methods for detailed

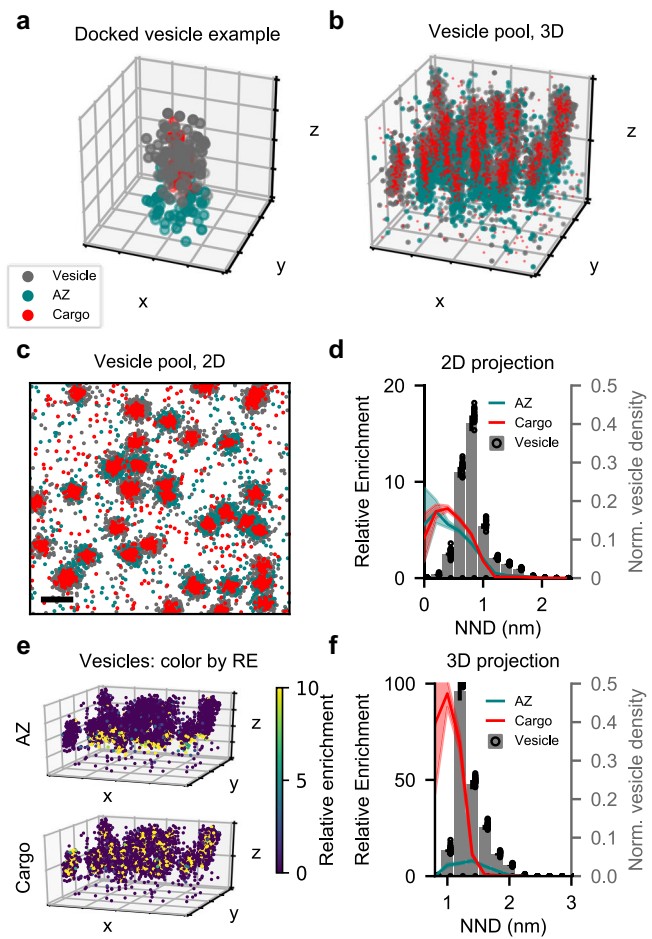

**Fig. 4 | Validation of RE-scoring in 3D. a** Simulation of a single docked vesicle with three molecular species: a vesicle membrane-bound, a SNARE-like and a lumenal. Grid is 50 by 50 nm. **b** Representative simulation of twenty vesicles. Half docked at the membrane and associated with SNARE-like molecules, and half in the intracellular space without SNARE association. Grid is 200 by 200 nm. **c** 2D projection of representative simulation in (**b**). Scale bar is 100 nm. **d** Vesicle Voronoï regions binned by nearest neighbour distance in 2D, with mean RE value of each bin for both AZ and cargo as line plot. Shaded area indicates S.E.M., $n = 20$. Scale bar is 100 nm. **e** Simulated vesicle localizations from (**b**) color-coded for their relative enrichment of either SNARE (top) for lumenal (bottom) localizations. Grid is 200 by 200 nm. **f** Vesicle Voronoï regions binned by nearest neighbour distance in 3D, with mean RE value of each bin for both AZ and cargo as line plot. Shaded area indicates S.E.M., $n = 20$.

protocol). Under basal conditions, the two proteins appear to form nanoclusters (Fig. 3a, b). This is in line with findings showing that both proteins exist in clusters at resting conditions[24,29]. Next, we assessed colocalization with our proposed method. As relative enrichment can be examined bidirectionally, we alternated the appointed primary and reference species: colocalization of primary Stx1 to reference Munc-18 (Fig. 3c, left) or colocalization of primary Munc-18 to reference Stx1 (Fig. 3c, right). Both species showed a propensity to colocalize with the densest clusters of the opposing species, as maximal enrichment is at localizations with less than 10 nm on average to adjacent molecules of the same species (Fig. 3d, e). However, the relative enrichment was higher for Munc-18 on Stx1, possibly because Stx1 is in functional excess relative to Munc18-1, as Stx1 has multiple independent roles in neuronal maintenance and neurotransmitter release[20]. This relative over-enrichment of Munc18-1 on Stx1 as compared to vice-versa can also be illustrated by color-coding the reference species based on individual RE score, as previously shown (Fig. 3f). Here, we plot the individual localizations rather than the associated regions, as the

border between extracellular space and cells is harder to distinguish in these more complex neuronal structures when using Voronoï regions and to avoid visual domination by less dense regions (Fig. S3a). Next, we treated the primary hippocampal cultures with either N-methyl-D-aspartate (NMDA) to activate NMDA-type ionotropic glutamate receptors and stimulate vesicular neurotransmitter release, or with tetrodotoxin (TTX), a sodium channel blocker, to prevent vesicular release (Fig. 3g). As Munc18-1 was more tightly associated with Stx1 during basal conditions, we focused on this enrichment-directionality. Upon stimulating release with NMDA, enrichment of Munc18-1 on Stx1 dropped in the densest Stx1 localization (Fig. 3h), in line with their proposed role in the synaptic vesicle fusion[22]. In a mirror image, blocking synaptic vesicle release with TTX increased the propensity of the two proteins to colocalize, consistent with an increase in vesicular docking or priming. This effect can again be visualized by plotting the reference species, Stx1, color-coded by RE score (Fig. 3i, Fig. S3b). We observed no significant redistribution of Stx1 after treatment (Fig. S3c), matching previous findings when analyzed with tessellation[30]. But tessellation may not uncover all reorganizational nuances. Results from applying the same analysis with primary and reference species reversed is shown in Fig. S3d, e.

Lastly, we compared our results to the other tessellation-based method, Coloc-Tesseler[12]. This method calculates either Spearman or Mander's coefficient, and while both can be computed in two ways similarly to our analysis, they are not directly comparable. When comparing the coefficient values of NMDA and TTX treated cells (Fig. 3j), neither of the four coefficients showed a significant difference, in contrast to both our proposed method and literature findings.

## Validation on a simulated 2D and 3D vesicle example

As membranes are relatively flat on a mesoscopic level, 2D models of plasma membrane-bound proteins, and by extension 2D images, capture a good part of the relevant biology. Some molecules, however, reside in the cytoplasm or on the membrane of different organelles. To fully capture their spatial distribution, analyses need to incorporate z-positions from 3D images. The same is true in thicker tissue samples. To test the 3D capabilities of our method, we simulated vesicles at an active zone. Three different molecular species were simulated: a plasma membrane-bound active-zone-like (AZ) species, i.e. SNAP-25, a species bound to the vesicle membrane, i.e. synaptophysin, and a cargo species in the lumen of the vesicle, i.e. neuropeptides (Fig. 4a). As input to our analysis, we generated ten images with twenty vesicles in each, all of which contained the cargo molecule but only half were docked and associated with a cluster of the AZ molecule (Fig. 4b). As most 3D SMLM techniques have a higher uncertainty in the z-plane, the random jitter representing technical uncertainty was doubled for this axis. To mimic the results that similar biology would yield with 2D SMLM, we projected the simulated 3D images on to a 2D plane (Fig. 4c).

When assessed in 2D, using the vesicle species as reference, the AZ and cargo species have equal RE score across the vesicle density distribution (Fig. 4d). They both appeared as highly enriched around the densest vesicle membrane localizations and fell below an RE score of 1 outside of the vesicles, indicating under-enrichment in the surrounding space. In contrast, a different distribution was resolved when analyzed in 3D (Fig. 4e). Here, a high degree of colocalization between the cargo and vesicles species was still evident, whereas the enrichment of the AZ species was reduced as expected, as apparent from the RE score (Fig. 4f). Based on the histogram of vesicle NND, we can see a preferential colocalization of the cargo to the dense-most localizations of the vesicle, whereas the AZ species peaks at the intermediate density, suggesting a more peripheral association. This becomes evident when plotting the vesicle localization color-coded for their RE score of the two species

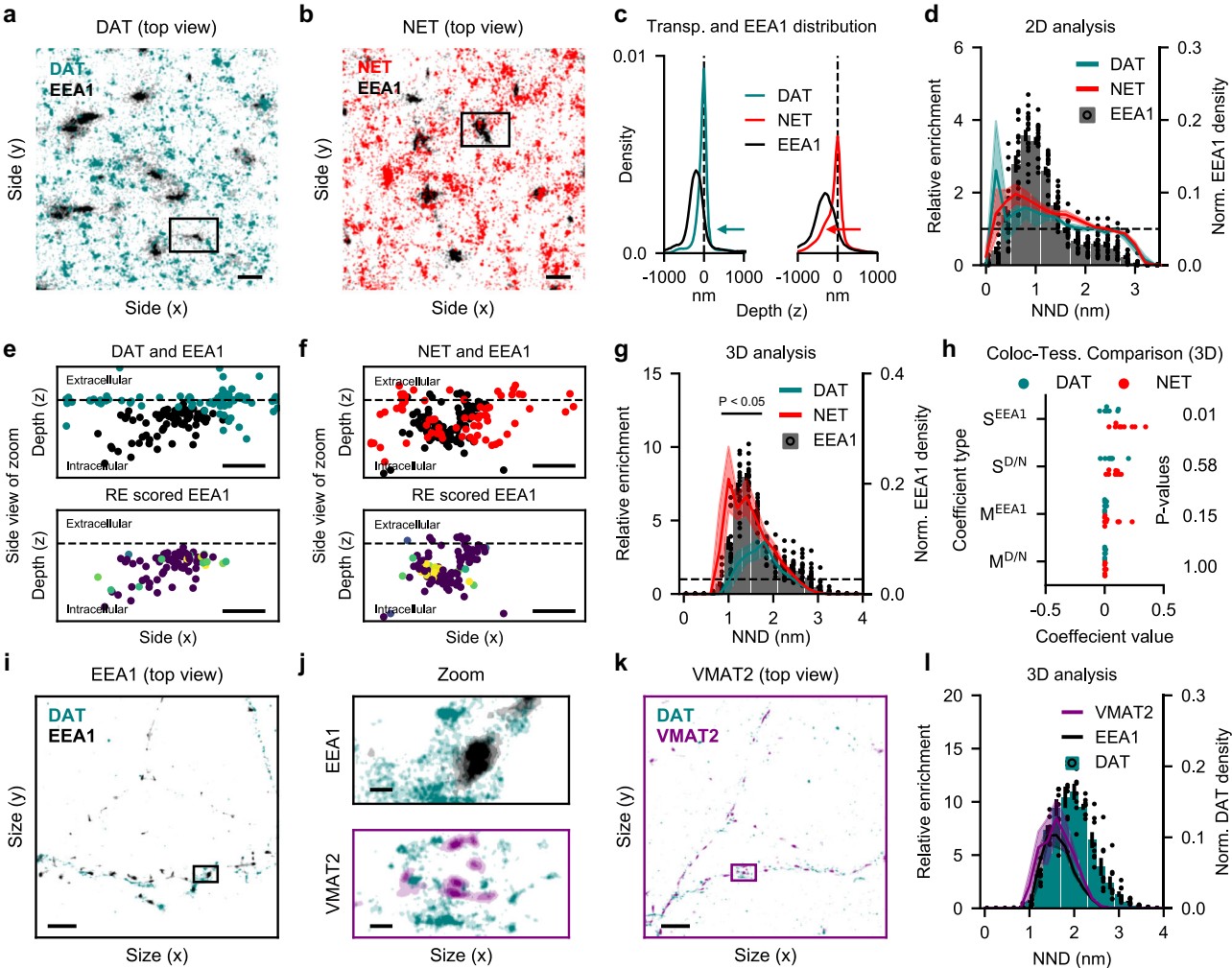

**Fig. 5 | Neurotransmitter transporter localization to early endosomes.**
**a** Representative top view of image with EEA1 and DAT in PC-12 cells acquired with astigmatic dSTORM. Molecular localizations visualized by gaussian representation. Scale bar is 0.5 μm. **b** Representative top view of image with EEA1 and NET in PC-12 cells acquired with astigmatic dSTORM. Molecular localizations visualized by gaussian representation. Scale bar is 0.5 μm. **c** Mean distribution of transporter and EEA1-positive structure depth. Normalized to peak transporter depth. Arrows indicate the differential intracellular accumulation of NET. **d** EEA1 Voronoï regions binned by nearest neighbour distance in 2D, with mean RE value of each bin for both DAT ($n = 7$ images) and NET ($n = 8$ images) as dots. Error bars area indicates S.E.M.. **e, f** Side view of black box from (**a**) and (**b**). Top panel shows both species, and bottom panel EEA1 localization color-coded by relative enrichment (RE) score. Dashed lines indicate plasma membrane depth as assessed by median transporter location in z. Scale bar is 200 nm. **g** Same as (**d**) but assessed in 3D ($H_0$: DAT = NET.

Black line indicates bins with $p < 0.05$. Unpaired, one-sided student's t-test by image). **h** Spearman (S) and Mander's (M) coefficient computed in 3D as per[12] ($H_0$: DAT = NET. $S^{EEA1}$, $p = 0.01$; $S^{D/N}$, $p = 0.58$; $M^{EEA1}$, $p = 0.15$; $M^{D/N}$, $p = 1.00$. Unpaired, one-sided student's t-test by image, FWER correction with Bonferroni-Holm). **i** Representative top view of image with EEA1 and DAT from primary midbrain cultures acquired with astigmatic dSTORM. Molecular localizations visualized by gaussian representation. Scale bar is 4 μm. **j** Zoom on boxes marked on (**i**) and (**k**) showing representative structures. Scalebars are 200 nm. **k** Representative top view of image with VMAT2 and DAT from primary midbrain cultures. Scale bar is 4 μm. **l** Relative enrichment assessed with DAT as reference species and either VMAT2 or EEA1 as the primary. Error bars and shaded area indicate S.E.M. ($H_0$: VMAT2 = EEA1, unpaired, two-sided student's t-test, $n = 4$ cover slips for both conditions.) Source data are provided as a Source Data file.

(Fig. 4e). The AZ species clearly enriches the bottom portion of vesicles near the membrane, whereas the cargo species highlights the center of all vesicles. This difference is much clearer in 3D as compared to the 2D visualization (Fig. S4a, b). Analysis of these simulated data underscores the importance of a 3D-compatible colocalization analysis for certain biological contexts.

While 3D data provides a greater amount of information than 2D, the extra dimension requires significantly more processing power. However, the relative enrichment function has acceptable processing times as compared to Coloc-Tesseler[12]. For a two-color 3D SMLM image with half a million localization in each channel, loading and processing took on the order of 30 to 60 seconds for the relative enrichment function across the computers we tested, whereas Coloc-Tesseler for comparison finished in 5 to 10 seconds.

## Investigating protein trafficking itineraries
Protein trafficking is an important element in cell biology, where 3D information is often needed to draw a conclusion. To investigate whether our method can inform us of trafficking itineraries, we tested it on experimental data of endocytosis of two different transmembrane proteins, the dopamine (DAT) and noradrenaline transporter (NET). The two transporters mediate clearance of their respective substrates, dopamine and noradrenaline, and play an important role in shaping the spatiotemporal profile of dopaminergic and noradrenergic neurotransmission[31,32]. Interestingly, DAT and NET display differences in subcellular localization and trafficking itinerary[30,33–35]. Specifically, NET has been reported to internalize more rapidly than DAT and preferentially sort to the Rab11 recycling compartment rather than for degradation, whereas the internalized DAT sort less to this

compartment. However, the specific path for the two transporters through near-membrane organelles is difficult to assess with conventional LM, due to their proximity to the cell surface. A central compartment to this near-surface trafficking is the early endosome. These amorphic structures are typically 400-700 nm in diameter[36], and play a key role in membrane-protein sorting following their internalization[37]. To test if the difference in internalization rates of the two transporters is reflected in the early endosomal steady-state content, we acquired dual-color dSTORM images of the early endosome antigen 1 (EEA1), alongside either anti-DAT or -NET antibodies in PC12 cells that express both transporters. Images for both transporters were captured with astigmatic dSTORM to achieve 3D resolution.

Visualizing the images in 2D from bottom-up by a Gaussian representation of the data, larger EEA1-positive formless structures with various protrusions are visible, matching the size described in the literature (Fig. 5a, b). Likewise, DAT clusters match what has previously been described[7], and NET architecture resembles that of DAT (Fig. 5a, b). Additionally, when assessing depth (z-axis position) of transporters and EEA1-positive structures (Fig. 5c), an intracellular accumulation is present for NET, but not for DAT, matching the expected distribution. We first assessed if a differential colocalization pattern to EEA1 could be resolved for NET and DAT in 2D images. This was done by applying our method without depth (z-coordinates) (Fig. 5d). We used EEA1 as the reference species, and while both transporters showed slight enrichment in the denser endosome localizations, indicating some association, there was no difference between the relative enrichment of NET and DAT in EEA1-positive structures. To assess if 3D information changes the result, we included depth information in the images (Fig. 5e, f, upper panels). Defining plasma membrane depth as the median transporter position in the z-axis, EEA1-positive structures were primarily intracellular, but proximal to the surface (Fig. 5c). Additionally, a qualitative assessment indicated more NET in these structures. When plotting EEA1 localization color-coded for their RE-values, the association of DAT and NET to intracellular EEA1 became both qualitatively visible and quantifiable, particularly for NET (Fig. 5e, f, lower panels). We reassessed relative enrichment by bins in 3D, and from the plot it becomes apparent the densest EEA1 localizations, the putative early endosomes, were significantly more enriched with NET than DAT (Fig. 5g). This provides clear evidence that the higher steady-state internalization rate of NET is, at least in part, mediated by sorting through early endosomes. Utility of the relative enrichment method is underscored when we compare our results to conventional colocalization measures as analyzed by Coloc-Tesseler (Fig. 5h, Fig. S4c). Indeed, only one of four measures showed significant difference in the association of the two transporter species to EEA1. Moreover, the single colocalization values provided by these methods offered no context, compared to the quantitative output and the qualitative visualization of our method.

Finally, we wanted to test our relative enrichment method on structures smaller than early endosomes. We therefore turned to midbrain cultures of dopaminergic neurons. These neurons express the vesicular monoamine transporter-2 (VMAT2) on both small synaptic vesicles and large dense-core vesicles[38]. Interestingly, previous data have indicated that, to facilitate loading of the vesicles with dopamine, VMAT2-positive vesicles might tether to DAT through protein-protein interactions with synaptogyrin-3[39]. This would implicate possible co-localization of the two proteins, and to test this possibility, we imaged DAT together with either VMAT2 or, as a reference molecule, EEA1. The appearance of EEA1 was equivalent to what we observed in PC12 cells (Fig. 5i, j). For both DAT and VMAT2 (Fig. 5i–k), we observed a clearly clustered signal in agreement with our previous findings[7]. The VMAT2 clusters ranged in size from 40 to 200 nm in diameter, matching well with reported sizes of synaptic vesicles and dense-core vesicles[38]. For the co-localization analysis we isolated dopaminergic neurons by DAT expression and used the identified ROI

as total area in the RE calculations (Fig. S5c-e). While we observed little DAT in early endosomes in PC12 cells, EEA1 was more enriched in the upper ranges of DAT densities in these dopaminergic neurons (Fig. 5l). This is likely explained by early endosomes being confined to terminals in dopaminergic neurons where DAT expression is at its highest[34,40]. Notably, VMAT2 showed an enrichment across DAT densities equal to EEA1, suggesting little specific tethering to DAT in primary midbrain cultures.

## Discussion

Colocalization analysis is an import tool in LM. Knowing the spatial distribution of a molecule relative to other molecules or compartments in the cell may help infer its function[1]. Indeed, the advent of super-resolution microscopy techniques has provided an even greater spatial resolution for this approach. However, analytical challenges have emerged with these new data. For instance, most of the newly developed methods for colocalization of coordinate-based molecular positions require user-defined input parameters that influence results, and whose impact require in-depth knowledge of the tool to fully grasp[8–11]. Levet and colleagues address that in their two-way tessellation-based approach[12], but like most colocalization analysis tools the final output is reduced to a single value, providing no context or visual representation.

In this work, we have presented an alternative approach to colocalization analysis. Here the co-organization of two molecular species is assessed in a density context relevant to a wealth of emerging biological questions, as heterogenous organization of plasma membrane as well as cytoplasmic proteins has become apparent. A parameter-free computation of the RE score provides the fold-above-expected occurrence of a primary species, across the density distribution of a reference species. We recommend binning by average distance to nearest neighbor, rather than by region size, as the distance is more easily interpretable than area or volume. The method is bilaterally applicable, as the species can be freely interchanged based on the biological question. It is worth mentioning that interpretation of the absolute RE score can be more complicated in 3D analyses - particularly when molecules are restricted to certain structures. This is the case for transporter localization to early endosomes or vesicles, where neither NET nor DAT have free roam of the cytoplasm and must reside in the membrane of organelles. To use the absolute RE values in instances like these, careful thought must be put into the biology. In addition to the density context provided by the RE plot, the scoring of each individual reference localization allows for semi-quantitative visualization of the co-organization that can aid deciphering of the underlying biology. The raw data can be visualized in color-coding based on RE score, with either polygon plots or scatter plots, as shown throughout the study. This visualization tool is a strong hybrid between qualitative and quantitative approaches, that relies on the scoring of each individual localization. It is inspired by Malkusch and colleagues[9], and distinguished the method from ones that are Mander's and Spearman's coefficient-based, where scoring is derived macroscopically. For simplicity and to avoid user-determined parameters, we evaluate enrichment based on density. However, as it may be prudent in certain biological contexts to separate nanodomains based on size, the Python script provides all tessellated regions as an output. From these users can segregate clusters as the biology warrants. Further, Voronoï tessellation is not restricted to three dimensions, potentially allowing the addition of a temporal dimension if used in conjunction with live-cell SMLM. As with any post-processing analysis tool for SMLM it is vital to correct for multiple blinking in the pre-processing to avoid a skewed result[41,42].

The algorithm is available in Python, rather than as a stand-alone OS-specific software, for easy implementation in an analysis pipeline and future open development. This comes with the added benefit that more and more SMLM-related pre- and postprocessing tools are

Python-based[43–48]. It is compatible with both 2D and 3D images, and existing Python packages allow for the loading of most all conceivable data formats.

## Methods

### Ethical Statement
The experiments were conducted in accordance with the Animal Experimentation Inspectorate, Denmark. All efforts were made to minimize animal suffering and to reduce the number of animals used.

### Relative Enrichment Function
First step of calculating relative enrichment (RE) is selecting a reference and a primary species. The reference species then segments the total image ($I$) into regions by Voronoï tessellation, where $R$ denote the full set of indices belonging to the elected reference species and $(L_r)_{r \in R}$ is each individual localization in the set $R$. Each $L_r$ segments the image into a Voronoï region, $V_r$, whose region is the set of point in the image closer to $L_r$ than any other localization, $L_j$, in $R$. All regions with an infinite area (edge cases) and the primary localizations they contain are discarded. The RE of the remaining Voronoï regions ($V_r$) are then computed by dividing observed occurrences of the primary species ($(L_p)_{p \in P}$) inside the region, where $P$ denotes the full set of indices belonging to the elected primary species, with the number of expected occurrences, $RE = \frac{L_p^{obs.}}{L_p^{expc.}}$ (Eq. 1). The number of expected occurrences is calculated by dividing the area of the region with the total area analyzed and multiplying it by total number of primary localizations, $L_p^{expc.} = \frac{V_r^{area}}{\sum V_r^{area}} |L_r|$ (Eq. 2).

### Primary hippocampal cultures
Hippocampal neurons from E19 (embryonic) Wistar rat embryos of mixed gender (Charles River, Wilmington, MA) were plated on 15 mm coverslips using 6 well plates in at a density of approximately 100.000 cells/well. The coverslips were sterilized with UV and first coated with poly-L-lysine (Gibco) overnight and then coated with Neurobasal media (Gibco) supplemented with 4% FBS (Gibco), 1:100 Glutamax (Gibco), 100 U/mL penicillin, and 10 mg/Ml streptomycin (Invitrogen) overnight. The rat brains were removed and placed in ice-cold dissection media: HBSS (Gibco) supplemented with 30 mM glucose, 10 mM HEPES (Gibco) (pH 7.4), 1 mM sodium pyruvate (Gibco), 100 U/mL penicillin, and 10 mg/mL streptomycin (Sigma). The brains were cut in half by sagittal incision. To access the hippocampus, each hemisphere has the medial section facing up and the corpus callosum was removed. After isolation, the hippocampi were treated with sterile filtered papain solution (Worthington) at 37 C for 20 min, triturated with two different diameter fire-polished Pasteur pipettes 10 times each, and filtered through a 70 μm cell strainer to remove cell debris. Density was calculated by a hemocytometer. The cells were seeded in Neurobasal media supplemented with 4% FBS, 2% B27 supplement (Gibco), 1:100 Glutamax (Gibco), 100 U/mL penicillin, and 10 mg/ml streptomycin (Invitrogen). After 24 h of seeding, the growth media was substituted with serum and glutamate-free media and cultured for 12–14 days in vitro (DIV). The cultures were stored at 35 C with 5% CO2. Media was changed every 3-4 days by replenishing half the media with fresh growth media.

### Primary midbrain cultures
Primary midbrain cultures were prepared essentially as described by Rahbek-Clemmensen and colleagues[7]. Briefly, the ventral tegmental area and substantia nigra were surgically isolated from P1 to P3 Wistar rats (Charles River, Germany) and treated with papain (116 mM NaCl, 5.4 mM KCl, 26 mM NaHCO3, 2 mM NAH2PO4, 1 mM MgSO4, EDTA, 25 mM glucose 1 mM cysteine, 0.5 mM Kyrunate, and Papain 20 U ml$^{-1}$) at 37 °C for 25 min. Next, we generated a single cell suspension by triturating gently, followed by spinning at 100 x g for 10 min and seeding in heated SF1C [50% (v/v) modified Eagle's medium (MEM),

40% (v/v) Dulbecco's modified eagles medium (DMEM), 10% (v/v) F-12 (all from Invitrogen) supplemented with 1% (v/v) heat-inactivated calf serum (FBS, 2.5 mg ml$^{-1}$, Invitrogen), 0.35% (w/v) D-glucose, 0.5 mM glutamine, 5 mM Kyrunic acid, Penicillin, Streptomycin, liquid catalse (0.05%), and DiPorzio61 (containing insulin, transferrin, superoxide dismutase, progesterone, cortisol, Na2SeO3, and T3)] on 1 M KOH sonicated coverslips (160-180 μm in thickness) coated with Poly-D-Lysine and pre-seeded with a glia cell monolayer. A few hours after seeding we added glia-derived neurotrophic factor. Cell proliferation was inhibited by adding 5-Fluorodeoxyurdine after 6-7 days. The cultures were imaged after 8–16 DIV.

### Heterologous cell cultures
PC-12 cells are derived from neuroendocrine chromaffin cells, and were maintained in Dulbecco's Modified Eagle Medium with 10% fetal bovine serum, 5% horse serum and 1% penicillin/streptomycin.

### Transfection
PC-12 cells were transfected in T25 flasks the after seeding 1 mio. cells in fresh media. 3 μl of lipofectamine was added to 100 μl of Opti-MEM and 1 μl of 1 μg/μl of DNA to another 100 μl of Opti-MEM. The lipofectamine mixture was left for 5 min. to form micelles and after another 25 min. with the DNA-mixture added, the combined solution was added to the T25 flasks and left at 37 °C O/N. The following day the flasks were washed with PBS, cells dissociated with trypsin and split into six- or twelve-well plates with coverslips coated with poly-L-ornithine.

### Immunocytochemistry
The cell samples were fixed in paraformaldehyde (3%) and washed three times in glycine (20 mM) and NH4Cl (50 mM) in PBS. Subsequently, the cells were washed in blocking buffer (5% Donkey serum, 1% BSA in PBS), and incubated in blocking permeabilization buffer (blocking buffer with saponin (0.2%)). Primary antibody was applied in blocking buffer for 60 min, followed by 3–5 min incubation in blocking buffer. Secondary antibody was applied in blocking buffer for 45 min, and the sample was incubated 2x for 5 min in blocking buffer. Samples were washed in PBS twice and post-fixated in paraformaldehyde (3%) for 15 min. Samples were washed twice in glycine (20 mM) and NH4Cl (50 mM) in PBS and stored in PBS at 4 °C until imaging

### Antibodies
We used the following primary antibodies for staining: Anti-Stx1 at 1:500 (Synaptic Systems, 110 011), anti-Munc18-1 at 1:500 (Synaptic Systems, 116 002), anti-hDAT at 1:1000 (Sigma-Aldrich, MAB369), anti-hNET at 1:1000 (MAb Technologies, NET17-1), anti-EEA1 at 1:1000 (Abcam, Ab2900) and anti-VMAT2 at 1:4000 (kindly provided by Dr. Garry Miller, Columbia University, New York). Secondary antibodies with conjugated AF568 or AF647 were added in 1:400 dilution.

### dSTORM
For dSTORM we used a buffer containing β-mercaptoehtanol and an enzymatic oxygen scavenger system (10% (w/V) glucose, 1% (V/V) beta-mercaptoethanol, 2 mM cyclooctatetraene, 50 mM Tris-HCl (pH 8), 10 mM NaCl, 34 μg mL$^{-1}$ catalase, 28 μg mL$^{-1}$ glucose oxidase). The imaging was performed with an ECLIPSE Ti-E epifluoresence/TIRF microscope (NIKON, Japan) equipped with 405 nm, 488 nm 561, and 647 nm lasers (Coherent, California, USA). All lasers were individually shuttered and collected in a single fiber to the sample through a 1.49 NA, 100x, apochromat TIRF oil objective (NIKON). For dual-color dSTORM, we used a dichroic mirror with the range 350–412, 485–490, 558–564, and 637–660 nm (97,335 QUAD C-NSTORM C156921). The excitation light was filtered at the wavelengths: 401 ± 24 nm, 488 ± 15 nm, 561 ± 15 nm, 647 ± 24 nm. The emitted light was filtered at the wavelengths: 425–475, 505–545, 578–625, and 664–787 nm, and

secondly by an extra filter to decrease noise (561 nm Longpass, Edge Basic, F76-561, AHF). A motorized piezo stage controlled by a near-infrared light-adjusted perfect focus system (NIKON) was applied to the system to reduce any sample drift over time in the z-direction. Dual-color dSTORM images were constructed from 10,000 frames for each color, taken at a 16 ms frame rate, with each color alternating by frame. Photons were collected with an iXon3 897 EM-CCD camera (Andor, United Kingdom). Laser powers used were 2.3 kW cm$^2$ for 647 nm, 1.0 kW cm$^2$ for 488 nm and for 561 nm. The 405 nm laser was used to incrementally increase blinking behavior at power <0.1 kW cm$^2$. For 3D-dSTORM, a cylindrical lens was placed before the camera to impart astigmatism. A reference z-stack of fluorescent beads (TetraSpeck) was acquired with 25 nm intervals spanning 2 μm before each imaging session. Localizations from dSTORM videos were fitted with 3D-DAOSTORM[44], with a background sigma of 8, maximum likelihood estimation as the fitting error model, 20 peak identification iterations, an initial sigma estimate of 1.5 and a threshold adjusted to each imaging session. A linear fit in both the x-z and y-z plane was subtracted to account for coverslip tilt.

### Statistical analysis

Choice of statistical analysis is presented in the legends associated with each figure, and where specified multiple testing was corrected for using the Bonferroni-Holmes correction. All *n*-values are individual images or simulations. Statistical analyses were carried out in Python 3.6.10 with the open-source python packages SciPy v1.5.2, Numpy v1.18.1, and Seaborn v0.11.0, and linear models in Statsmodels v0.12.2. Boxplots show 25th and 75th percentile, with whiskers indicating data within 1.5 times the interquartile range. Remaining data were plotted as outliers. No statistical methods were used to predetermine sample sizes.

### Reporting summary

Further information on research design is available in the Nature Research Reporting Summary linked to this article.

## Data availability

The processed data generated in this study are provided in the Source Data file. Due to size the raw dSTORM images are not included, but available from the corresponding author within two weeks upon request. Source data are provided with this paper.

## Code availability

All custom python functions and working examples with data from the simulated 2D and 3D vesicle example are available online at: https://github.com/Ejdrup/relative-enrichment/releases/tag/v0.1.0 or at DOI: 10.5281/zenodo.6627703. The figure-specific analyses are available from the corresponding author within two weeks upon request.

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

## Acknowledgements
The work was supported by the Lundbeck Foundation grants R266-2017-4331 (UG), R276-2018-792 (UG), R230-2016-3154 (M.D.L.) and R303-2018-3540 (F.H.).

## Author contributions
A.L.E. developed the method with help from M.D.L. and N.L. Experiments were performed by A.L.E., N.L., M.D.L. and A.K.. Simulations and analyses were carried out by A.L.E. A.L.E. wrote the manuscript. A.L.E., M.D.L., F.H.H., K.L.M. and U.G. edited the manuscript. All authors reviewed and critically evaluated the manuscript. K.L.M. and U.G. supervised the project.

## Competing interests
The authors declare no competing interests.
