## [Peer Review File · Nature Communications]

Reviewers' Comments:

Reviewer #1:

Remarks to the Author:

The authors describe the novel concept of "relative enrichment" (RE) in the context of two-colour single-molecule localisation microscopy, based on the density of detections of molecule A in the "reference" Voronoi tiles of molecule B (and vice versa). The approach described is mathematically simple and elegant, essentially parameterless, works in 2D and 3D, and has a particular strength in that rather than just returning a single value as an indicator of co-localisation (as done by Coloc-Tesseler in particular), RE allows assessment of co-localisation relative to detection density via direct visualisation of colour-coded Voronoi diagrams and plots of enrichment against histograms of Voronoi tile area or nearest neighbour distance.

The authors initially used synthetic 2D two-colour data of various geometries to demonstrate its utility; then 2D two-colour neuronal data for syntaxin and Munc-18, using RE to demonstrate that treatment of neurons with N-methyl-D-aspartate or tetrodotoxin significantly altered the co-localisation.

They then apply it to synthetic 3D data representing docked synaptic vesicles and associated active zone, vesicular and luminal proteins, which highlights the ability of 3D analysis to define co-clustering relationships that are masked in 2D. As a climax to the paper, the authors show that 3D EA was able to resolve differences in the 3D co-localisation of dopamine transporter (DAT) or noradrenaline transporter (NET) with early endosome antigen 1 (EEA1) in PC12 cells, indicative of underlying differences in the early endosomal transport of these proteins. Similarly, to the synthetic 3D study, these differences were masked in 2D.

The article was a pleasure to read - its central premise is clear and concisely explained without unnecessary mathematics, it is very well written, and the data is presented in a clear, visually appealing and easily digested manner. RE's advantages over other established techniques are explained without excess grandiosity in the text and made evident in the figures. The interpretation of and biological conclusions reached from RE analysis of the neuronal and PC12 data are sound.

This is an interesting technique although a number of essential issues should be addressed:

Major:

1- As it stands, there is no way to quantitatively compare the size of the clusters and the level of colocalization within each of the markers considered (degree of cluster overlap etc.). This is an important point because the level of colocalization could vary in and out of clusters and even within clusters. Further, this would be essential to quantitatively compare the data generated using this technique and that of others. This should be included in both the in silico and in the syntaxin Munc18-1 dataset. This way they'll be able to benchmark their results with the cluster size and clusters.

2- The last dataset on EEA1 is interesting although early endosomes are fairly large structures and it would be nice if the authors could challenge their techniques with synaptic vesicles which are smaller and pack inside the presynapses.

3- Literature: Munc18 and syntaxin cluster have been discussed by a number of groups whose work has not been fully cited including Duncan, Lang and Meunier. This is especially important with regard to point 1 (benchmarking).

Minor:

4- Some comparison on the speed of RE vs Coloc-Tesseler could be included given that throughput is often a consideration when adopting new techniques.

5- The authors have (wisely, given the ascendancy of Python) chosen to implement RE via an easy to implement Python module whose output can be further manipulated and visualised by the user. The instructions on the GitHub repository, as well as example code, are clear and useful. The average moderately computer savvy SMLM researcher should have no problem downloading and running the code. Given that one of the scripts is essentially a recapitulation of the process used in the 3D synthetic data portion of the manuscript, I would consider the findings reproducible. However, the power of RE would be more readily and widely usable in the future if the authors would consider packaging it in a GUI - this reviewer highly recommends PySimpleGui as a way to

rapidly wrap arcane super resolution microscopy tools into standalone user-friendly graphical interfaces operable by non-coders.

6- Lastly, while RE is clearly a powerful tool for qualitative and quantitative spatial co-localisation analysis, the temporal aspects of co-localisation are not considered/explored i.e. are molecules co-localising in both space and time. Could the technique feasibly be expanded to encompass this for analysis of live cell SMLM data? eg 4D [x,y,z,t] Voronoi segmentation? Perhaps some discussion of this would be germane. Whilst on the subject of live cells (trajectory centroids or all detections) vs fixed cell SMLM data, some discussion of the utility of RE in both of these contexts would be useful.

7- I found a few typos in "coefficients" in some of the figures and the use of "luminal" instead of "lumenal" when referring to protein in the ER lumen.

Reviewer #2:

Remarks to the Author:

This article describes an interesting and potentially useful analysis method for comparing local densities of two molecular species in localization microscopy, using the density of one to set the scale for measuring the density of the other. The work is well-executed, well-described, and no doubt useful for various investigations. It belongs somewhere in the literature as a nice way to present data. I could quibble with phrasing in a few places (e.g. lines 73-76 made me stumble at first).

However, while it's a nice way to present data, I'm not sure that it's a high-significance advance. Computing densities in the cells of a Voronoi tessellation is a nice way to do things, but when I compare the images of localizations with the relative enrichment graphs, it isn't clear just how much more I've learned. More precise data presentations add value, but the magnitude of the value added here is unclear. Histograms of areas and neighbor distances are not major leaps.

Without a doubt this work belongs in a methods section, and probably in a protocols journal, but it isn't adding enough value to count as a major advance.

I hate being this negative because it's well-done work. Some well-done work is necessary and laudable but still not major.

Reviewer #3:

Remarks to the Author:

Here, Ejdrup et al present an algorithm for quantifying co-localisation in single-molecule microscopy data. The method takes as a starting point the published Voronoi idea of Levet et al and then counts how many points from a 2nd channel fall into each Voronoi cell. In general, I see that the method works and this is a measure of co-localisation, however I don't feel it is of sufficient novelty for Nature Communications. As the authors themselves point out there are numerous co-localisation methods for this kind of data, some of which even share methodological similarities with the ideas presented here. Ultimately, this method has minimal advantages over what already exists and doesn't generate any new capabilities for biological researchers.

On top of this, there are a number of technical issues that need to be addressed:

The visual output (e.g. Fig1E) is misleading to the eye. The colour is fine, but the most important areas – where the reference distribution is dense give small Voronoi cells and so are de-emphasised. Larger cells (which presumably are less interesting because there is no reference points there anyway) are large and appear as the most prominent.

One issue is assigning significance. A random distribution can produce enrichment by chance. How can the authors assign some p-value to whether enrichment is "real" and not the result of this chance?

There is insufficient testing on simulated data e.g. vs two random distributions (inc. varying the

density), both random but with individual points colocalising (i.e. the identical random distribution in each channel), clustered reference vs random primary and vice versa, both distributions clustered but with positive, negative and neutral co-clustering – all done with high 30-100 n numbers of regions etc.

No account seems to have been made for fluorophore multiple blinking. Small clusters of localisations in each channel caused by this could affect the co-localisation measure. Similarly, no account is taken of the localisation precision of each point meaning that badly localised primary points could end up in neighbouring reference Voronoi cells.

I assume it's right but the text isn't clear, but in calculating the number of expected primary localisations (line 86), the total area of all localisation is smaller because edge regions are deleted so the total number of observed primary localisations need to not include those in these border regions.

Reviewer #4:

None

RESPONSE TO REVIEWERS

(NCOMMS-22-03147-T Ejdrup et al., Relative enrichment – a density-based colocalization measure for single-molecule localization microscopy).

Reviewer #1

“The authors describe the novel concept of "relative enrichment" (RE) in the context of two-colour single-molecule localisation microscopy, based on the density of detections of molecule A in the "reference" Voronoi tiles of molecule B (and vice versa). The approach described is mathematically simple and elegant, essentially parameterless, works in 2D and 3D, and has a particular strength in that rather than just returning a single value as an indicator of co-localisation (as done by Coloc-Tesseler in particular), RE allows assessment of co-localisation relative to detection density via direct visualisation of colour-coded Voronoi diagrams and plots of enrichment against histograms of Voronoi tile area or nearest neighbour distance.

The authors initially used synthetic 2D two-colour data of various geometries to demonstrate its utility; then 2D two-colour neuronal data for syntaxin and Munc-18, using RE to demonstrate that treatment of neurons with N-methyl-D-aspartate or tetrodotoxin significantly altered the co-localisation.

They then apply it to synthetic 3D data representing docked synaptic vesicles and associated active zone, vesicular and luminal proteins, which highlights the ability of 3D analysis to define co-clustering relationships that are masked in 2D. As a climax to the paper, the authors show that 3D EA was able to resolve differences in the 3D co-localisation of dopamine transporter (DAT) or noradrenaline transporter (NET) with early endosome antigen 1 (EEA1) in PC12 cells, indicative of underlying differences in the early endosomal transport of these proteins. Similarly, to the synthetic 3D study, these differences were masked in 2D.

The article was a pleasure to read - its central premise is clear and concisely explained without unnecessary mathematics, it is very well written, and the data is presented in a clear, visually appealing and easily digested manner. RE's advantages over other established techniques are explained without excess grandiosity in the text and made evident in the figures. The interpretation of and biological conclusions reached from RE analysis of the neuronal and PC12 data are sound.”

We are thankful of the high praise of both content and manuscript presentation from this reviewer, as well as the very constructive criticism raised in both major and minor points. Below are our comments for each point.

**“This is an interesting technique although a number of essential issues should be addressed:
Major:**

1- As it stands, there is no way to quantitatively compare the size of the clusters and the level of colocalization within each of the markers considered (degree of cluster overlap etc.). This is an important point because the level of colocalization could vary in and out of clusters and even within clusters. Further, this would be essential to quantitatively compare the data generated using this technique and that of others. This should be included in both the in silico

and in the syntaxin Munc18-1 dataset. This way they'll be able to benchmark their results with the cluster size and clusters."

The reviewer correctly points out that our method quantifies cluster density rather than size. This is a valid concern, as both can be important metrics for cluster description, and in certain instance size might be an important biological feature. During development of the relative enrichment method, we discussed adding both a density and cluster size axis but decided against a 2-dimensional spatial description, as these plots were significantly harder to read and interpret. We focused on density over size as density has the distinct advantage that it does not require a manually set threshold to determine cluster cut-off. Levet and colleagues pre-define this cut-off at median area of tessellated regions, but it is nevertheless a manual parameter and may not represent an accurate reflection in all cases. Choosing density keeps the method parameter free with less manual user-input.

Cluster size may still be of importance to some researchers, however, and based on this reviewer comment we realize that our manuscript had omitted this aspect. Fortunately, we have written the first function of our code to provide all tessellated regions as an output, and users can manually sort for specific cluster sizes from these data. To underline the elements brought up by the reviewer and discussed here, we have added a section to the discussion touching on cluster size aspect at lines 333-337.

"2 - The last dataset on EEA1 is interesting although early endosomes are fairly large structures and it would be nice if the authors could challenge their techniques with synaptic vesicles which are smaller and pack inside the presynapses."

Upon this suggestion from the reviewer, we decided to incorporate a challenge of our technique with synaptic vesicles into the part of our study that focus on neurotransmitter transporters/EEA1. In primary neuronal cultures derived from the rat midbrain, we imaged DAT alongside either EEA1 or VMAT2, the vesicular monoamine transporter in dopaminergic neurons (see new Fig. 5I-K). We did this to test if we could detect VMAT2-positive vesicle-association to DAT through the reported synaptogyrin-3 interaction from Egaña et al., J. Neurosci. (2009).

In agreement with our previous findings (Rahbek-Clemmensen et al., Nature Communication 2017), we observed a clustered VMAT2 signal with clusters ranging from 40 to 120 nm in diameter, conceivably representing the presence of VMAT2 on both small synaptic vesicles and on large dense core vesicles (see new Supplementary Fig. 5A, B) in accordance with findings using electron microscopy by Wildenberg et al., eLife (2021). However, we found no difference in the association to DAT between VMAT2 or EEA1, whose appearance was equal to what we observed in PC12 cells (Fig. 5A, B, I and J). This indicates no specific association of VMAT2-vesicles to DAT, or an equal peripheral association of EEA1.

To isolate only DA neurons and account for the complex morphology of the neurons we used Otsu's thresholding (Fig. S5C-E). We applied the same technique to the PC12 cells, which significantly enhanced the difference between NET and DAT association to EEA1.

The new data are described in lines 283-300 and shown in new panels I-k of Fig 5.

“3 - Literature: Munc18 and syntaxin cluster have been discussed by a number of groups whose work has not been fully cited including Duncan, Lang and Meunier. This is especially important with regard to point 1 (benchmarking).”

We appreciate the literature overview of the reviewer and have accordingly added the following citations to the Munc18-1/Stx1 section:

Zilly et al., PLoS Biol. (2006).

Martin et al., J. Cell. Sci. (2013).

Padmanabhan et al., Neuropharmacology (2020).

“Minor:

4- Some comparison on the speed of RE vs Coloc-Tesseler could be included given that throughput is often a consideration when adopting new techniques.”

Excellent suggestion. We have compared processing speed of both methods for a single 3D image with roughly 1,000,000 localizations. Coloc-Tesseler takes on the order of 5 to 10 seconds, including loading and processing, whereas the relative enrichment function takes on the order of 30-60 seconds, depending on the processor.

We have implemented the information into the manuscript on lines 229-234.

We would like to note that loading and processing time are not the only factors to consider when assessing overall speed and ease, as discussed, in the next point below.

“5 - The authors have (wisely, given the ascendancy of Python) chosen to implement RE via an easy to implement Python module whose output can be further manipulated and visualised by the user. The instructions on the GitHub repository, as well as example code, are clear and useful. The average moderately computer savvy SMLM researcher should have no problem downloading and running the code. Given that one of the scripts is essentially a recapitulation of the process used in the 3D synthetic data portion of the manuscript, I would consider the findings reproducible. However, the power of RE would be more readily and widely usable in the future if the authors would consider packaging it in a GUI - this reviewer highly recommends PySimpleGui as a way to rapidly wrap arcane super resolution microscopy tools into standalone user-friendly graphical interfaces operable by non-coders.”

Another good suggestion by the reviewer. However, we believe the python function packaging allows for a much more rapid overall analysis pipeline with little to no manual steps, compared

to a more laborious standalone GUI. One does not rule out the other, and while a GUI could make it more accessible to some users, given the already advanced pre-processing involved in SMLMs data, the majority of target users should already be moderately computer savvy.

“6 - Lastly, while RE is clearly a powerful tool for qualitative and quantitative spatial co-localisation analysis, the temporal aspects of co-localisation are not considered/explored i.e. are molecules co-localising in both space and time. Could the technique feasibly be expanded to encompass this for analysis of live cell SMLM data? eg 4D [x,y,z,t] Voronoi segmentation? Perhaps some discussion of this would be germane. Whilst on the subject of live cells (trajectory centroids or all detections) vs fixed cell SMLM data, some discussion of the utility of RE in both of these contexts would be useful.”

The reviewer brings up an excellent point we had not ourselves considered. Voronoi segmentation in 4D should mathematically be possible, although a computationally faster implementation would likely be needed, given the already marked increase in computation time from 2D to 3D. Alternatively, the time series could be discretized and plotted as either multiple lines of enrichment (i.e. akin to the treatments in Fig. 3H) or as a quasi-continuous 2D plane of enrichment, with x- and y-axis representing density and time and z-axis representing relative enrichment.

We have added a similar, truncated discussion to the manuscript on lines 337-339.

“7 - I found a few typos in “coefficients” in some of the figures and the use of “luminal” instead of “lumenal” when referring to protein in the ER lumen.”

We thank the reviewer for the keen eye and have corrected all the mistake we could find.

Reviewer #2

“This article describes an interesting and potentially useful analysis method for comparing local densities of two molecular species in localization microscopy, using the density of one to set the scale for measuring the density of the other. The work is well-executed, well-described, and no doubt useful for various investigations. It belongs somewhere in the literature as a nice way to present data.”

We greatly appreciate the compliments of the work and the manuscript itself, as well as the shared belief in the usefulness of this work.

“ I could quibble with phrasing in a few places (e.g. lines 73-76 made me stumble at first).”

We agree with the reviewer and, therefore, we have rewritten the specific passage, as well as reformulated other “heavy” paragraphs in the manuscript (e.g. lines 279-282).

“However, while it's a nice way to present data, I'm not sure that it's a high-significance advance. Computing densities in the cells of a Voronoi tessellation is a nice way to do things, but when I compare the images of localizations with the relative enrichment graphs, it isn't clear just how much more I've learned. More precise data presentations add value, but the magnitude of the value added here is unclear. Histograms of areas and neighbor distances are not major leaps. Without a doubt this work belongs in a methods section, and probably in a protocols journal, but it isn't adding enough value to count as a major advance.”

We do not agree that our work “belongs in a methods section” or “a protocols journal”. We also disagree that “the magnitude of the value added here is unclear”. In fact, we would strongly argue that in the manuscript we demonstrate the strong value and general utility of the relative enrichment measure, and thereby that indeed our study represents a major advance.

It is important to bear in mind that although super-resolution imaging techniques have provided unprecedented possibilities for improving our insights into the molecular organization and dynamic regulation of cellular structures, the techniques have also required the development of quantitative analyses appropriate for these new data types. However, it has proven particularly difficult to develop suitable quantitative methods that takes advantage of the high resolution when determining the degree of co-localization of two different molecular species e.g., two proteins in a cell.

Here, we describe such a suitable method - the relative enrichment measure - that not only is easily applicable but also reframes the question of colocalization by providing a density-context relevant to multiple biological questions. Previous methods rely on user-defined values that impact outcome, do not work with 3D data or are impacted by local molecular density. Lev et al and coworkers (Nature Communications 2019) published a method addressing several of the issues, but as with the previous work, colocalization is reduced to a single value, which arguably is a remnant from pre-super resolution diffraction-limited light microscopy imaging.

Importantly, we provide several examples showing how the method can be used to address specific biological questions.

First, we perform experiments on hippocampal neurons where we investigate co-localization of two presynaptic key proteins, Munc18-1 and syntaxin-1 (Stx-1) (Fig. 3). While Stx1 is critical for vesicle docking and fusion Munc18-1 orchestrate the assembly of release machinery. By application of our new method, we can demonstrate how release stimulation leads dissociation of the two proteins, as reflected by a drop in enrichment of Munc18-1 on Stx1 in the densest Stx1 localizations and, conversely, how inhibition of release increased the propensity of the two proteins to colocalize (Fig. 3). The data provide direct support to earlier studies in

neuroendocrine and neuronal cells suggesting that Munc18-1 and Stx1 colocalize at docked vesicles and disperse following vesicular neurotransmitter release.

In the second series of experiments, we turn to PC12 cells to demonstrate how we can use the method to visualize and substantiate distinct trafficking itineraries for two membrane proteins playing a key role in controlling monoamine homeostasis, the dopamine transporter (DAT) and the norepinephrine transporter (NET) (Fig. 5A-H). Specifically, we investigate the specific path for the two transporters through near-membrane organelles (e.g., early endosomes), which has been very challenging with conventional light microscopy, due to the organelle's proximity to the cell surface. While 2D analysis of the dual-color dSTORM data did not reveal differences in enrichment of DAT or NET onto the early endosome marker EEA1, 3D analysis revealed that the putative early endosomes, were significantly more enriched with NET than DAT (Fig. 5G). This represents clear evidence that the higher steady-state internalization rate of NET is, at least in part, mediated by sorting through early endosomes. The unique advantage of our relative enrichment method was underscored by comparison to conventional colocalization measures as Coloc-Tesseler (Fig. 5H, Fig. S4C). Notably, only one of four measures showed significant difference in association the two transporter species to EEA1.

In the revised manuscript, we apply moreover the method to synaptic vesicles in dopaminergic neurons. As outlined in further details above in our response to reviewer 1, we assess putative co-localization of the vesicular monoamine transport (VMAT-2) with DAT to test the hypothesis that VMAT2-positive vesicles associate to DAT through the reported synaptogyrin-3 interaction from Egaña et al., *J. Neurosci.* (2009). Indeed, we observed a clustered VMAT2 signal, conceivably representing the presence of VMAT2 on both small synaptic vesicles and on large dense core vesicles (see new Supplementary Fig. 5A, B). However, by application of the relative enrichment method, we found no difference in the association to DAT between VMAT2 or EEA1 (see new Fig. 5I-K), indicating no specific association of VMAT2-vesicles to DAT, or an equal peripheral association of EEA1.

In summary, we are fully convinced that the relative enrichment method represents a significant advancement in quantification of single-molecule localization microscopy data that allows substantial in-depth analyses for a wide crowd of scientists. Indeed, we show that the method can be used for addressing specific biological questions and that it may be superior to current available techniques. It is also important to emphasize the fact that our method enables visualization of the molecular scale organization directly in the color-coded images. No other tessellation approach to colocalization provides this possibility. Furthermore, the histograms of relative enrichment complement these images, as they show the overall organization of the data and confirm that the given examples are indeed representative. Consequently, we would strongly argue that our study is well suited for a journal like *Nature Communications*.

Reviewer #3

We thank the reviewer for the excellent suggestions that, among other things, address important controls for the relative enrichment method.

“Here, Ejdrup et al present an algorithm for quantifying co-localisation in single-molecule microscopy data. The method takes as a starting point the published Voronoi idea of Levet et al and then counts how many points from a 2nd channel fall into each Voronoi cells. In general, I see that the method works and this is a measure of co-localisation, however I don’t feel it is of sufficient novelty for Nature Communications. As the authors themselves point out there are numerous co-localisation methods for this kind of data, some of which even share methodological similarities with the ideas presented here. Ultimately, this method has minimal advantages over what already exists and doesn’t generate any new capabilities for biological researchers.”

We disagree with the reviewer that the method has minimal advantages over what already exists. We believe that we provide several strong arguments and examples throughout the manuscript where we highlight the novel insight provided by this method and that the method indeed generate new capabilities for biological researchers.

As we highlight in our response to reviewer 1, it is important to bear in mind that although super-resolution imaging techniques have provided unprecedented possibilities for improving our insights into the molecular organization and dynamic regulation of cellular structures, the techniques have also required the development of quantitative analyses appropriate for these new data types. However, it has proven particularly difficult to develop suitable quantitative methods that takes advantage of the high resolution when determining the degree of co-localization of two different molecular species e.g., two proteins in a cell.

Here, we describe such a suitable method - the relative enrichment measure - that not only is easily applicable but also reframes the question of colocalization by providing a density-context relevant to multiple biological questions. Previous methods rely on user-defined values that impact outcome, do not work with 3D data or are impacted by local molecular density. It is correct that Levet and coworkers (Nature Communications 2019) published a method addressing several of the issues, but as with the previous work, colocalization is reduced to a single value, which arguably is a remnant from pre-super resolution diffraction-limited light microscopy imaging.

Importantly, we provide several examples showing how the method can be used to address specific biological questions.

First, we perform experiments on hippocampal neurons where we investigate co-localization of two presynaptic key proteins, Munc18-1 and syntaxin-1 (Stx-1) (Fig. 3). While Stx1 is critical for vesicle docking and fusion Munc18-1 orchestrate the assembly of release machinery. By application of our new method, we can demonstrate how release stimulation leads dissociation of the two proteins, as reflected by a drop in enrichment of Munc18-1 on Stx1 in the densest Stx1 localizations and, conversely, how inhibition of release increased the propensity of the two proteins to colocalize (Fig. 3). The data provide direct support to earlier studies in neuroendocrine and neuronal cells suggesting that Munc18-1 and Stx1 colocalize at docked vesicles and disperse following vesicular neurotransmitter release.

In the second series of experiments, we turn to PC12 cells to demonstrate how we can use the method to visualize and substantiate distinct trafficking itineraries for two membrane proteins playing a key role in controlling monoamine homeostasis, the dopamine transporter (DAT) and the norepinephrine transporter (NET) (Fig. 5A-H). Specifically, we investigate the specific path for the two transporters through near-membrane organelles (e.g., early endosomes), which has been very challenging with conventional light microscopy, due to the organelle's proximity to the cell surface. While 2D analysis of the dual-color dSTORM data did not reveal differences in enrichment of DAT or NET onto the early endosome marker EEA1, 3D analysis revealed that the putative early endosomes, were significantly more enriched with NET than DAT (Fig. 5G). This represents clear evidence that the higher steady-state internalization rate of NET is, at least in part, mediated by sorting through early endosomes. The unique advantage of our relative enrichment method was underscored by comparison to conventional colocalization measures as Coloc-Tesseler (Fig. 5H, Fig. S4C). Notably, only one of four measures showed significant difference in association the two transporter species to EEA1.

In the revised manuscript, we apply moreover the method to synaptic vesicles in dopaminergic neurons. As outlined in further details above in our response to reviewer 1, we assess putative co-localization of the vesicular monoamine transport (VMAT-2) with DAT to test the hypothesis that VMAT2-positive vesicles associate to DAT through the reported synaptogyrin-3 interaction from Egaña et al., J. Neurosci. (2009). Indeed, we observed a clustered VMAT2 signal, conceivably representing the presence of VMAT2 on both small synaptic vesicles and on large dense core vesicles (see new Supplementary Fig. 5A, B). However, by application of the relative enrichment method, we found no difference in the association to DAT between VMAT2 or EEA1 (see new Fig. 5I-K), indicating no specific association of VMAT2-vesicles to DAT, or an equal peripheral association of EEA1.

In summary, we are fully convinced that the relative enrichment method represents a significant advancement in quantification of single-molecule localization microscopy data that allows substantial in-depth analyses for a wide crowd of scientists. Indeed, we show that the method can be used for addressing specific biological questions and that it may be superior to current available techniques. It is also important to emphasize the fact that our method enables visualization of the molecular scale organization directly in the color-coded images. No other tessellation approach to colocalization provides this possibility. Furthermore, the histograms of relative enrichment complement these images, as they show the overall organization of the data and confirm that the given examples are indeed representative. Consequently, we would strongly argue that our study is well suited for a journal like Nature Communications.

“On top of this, there are a number of technical issues that need to be addressed:

The visual output (e.g. Fig1E) is misleading to the eye. The colour is fine, but the most important areas – where the reference distribution is dense give small Voronoi cells and so are de-emphasised. Larger cells (which presumably are less interesting because there is no reference points there anyway) are large and appear as the most prominent.”

The reviewer brings up a great point. For the same reason we plotted color-coded points rather than regions for the remainder of the manuscript (Fig. 3F and I, Fig. 4F and Fig. 5E and F). We discuss this on line 178-180 in the original manuscript, but motivated by the reviewers comment we have expanded and added a line about visual domination (line 177, new manuscript).

“One issue is assigning significance. A random distribution can produce enrichment by chance. How can the authors assign some p-value to whether enrichment is “real” and not the result of this chance?”

As the reviewer writes, there is a non-zero chance that any given reference region will be randomly enriched. However, as the points accumulate, the corresponding randomly under-enriched regions will even that out, and a mean enrichment can be computed. A very small number of points will have a higher chance of spurious skew in the mean, but as shown on Fig. 2G and H as well as Fig. S1A, even a relatively small sample from a random distribution will yield an enrichment of 1 independent of local densities. This is further addressed by the simulations performed in response to another comment from the reviewer, as detailed further down in the document.

“There is insufficient testing on simulated data e.g. vs two random distributions (inc. varying the density), both random but with individual points colocalising (i.e. the identical random distribution in each channel), clustered reference vs random primary and vice versa, both distributions clustered but with positive, negative and neutral co-clustering – all done with high 30-100 n numbers of regions etc.”

Great point. We have now simulated increasing densities of random distributions, randomly distributed clusters, anti-colocalizing clusters and two perfectly colocalizing populations with identical but random over-all distributions (see new Fig. S2A-F). The results of the simulation are described in the manuscript on lines 131-141.

All three increasing densities of random distributions show the same result: a relative enrichment of 1 across densities apart from the extremes, where spurious skews can occur due to the low count.

The randomly distributed clusters result in the same RE score of 1, whereas the anti-colocalizing clusters show an RE value below 1 in the densest distributions and an enrichment outside of the clusters.

The identical random distribution by design yields an RE curve inverse to the mean area of the density bin, as this is an indicator of how likely a localization within the given region is. The relative enrichment function does not take into account how close to the seed of the reference region a given primary localization is.

“No account seems to have been made for fluorophore multiple blinking. Small clusters of localisations in each channel caused by this could affect the co-localisation measure. Similarly, no account is taken of the localisation precision of each point meaning that badly localised primary points could end up in neighbouring reference Voronoi cells.”

Multiple fluorophore-blinking and localization precision are definitely important to consider when performing SMLM experiments. However, these are ubiquitous issues and should be addressed in the pre-processing of all SMLM data sets. Relative enrichment is not an attempt to address pre-processing, but rather designed as a post-processing analytical tool. As for all experiments, proper controls are needed to account for technical issues and draw strong biological conclusions no matter the analysis applied.

“I assume it’s right but the text isn’t clear, but in calculating the number of expected primary localisations (line 86), the total area of all localisation is smaller because edge regions are deleted so the total number of observed primary localisations need to not include those in these border regions.”

Great point. We have implemented that information in the method section of the manuscript.

Reviewers' Comments:

Reviewer #1:

Remarks to the Author:

The authors have answered my queries and criticisms satisfactorily. And indeed, whether size matters or not is a question of perspective... They should carefully checked the new references added as not all of them are correct.

Reviewer #2:

Remarks to the Author:

This manuscript has been somewhat expanded since the first submission. Between the new material and the fresh perspective of reading again later, I concede that this manuscript presents a significant methodological improvement over existing tools. The parameter-free nature, and robustness of results against swapping between reference molecule and molecule of interest, really is something significant. I particularly like that the authors are able to very clearly establish the expected quantitative trends for perfect co-localization, complete independence, and anti-correlation, and show it in multiple sets of real data. I recommend publication with only minor edits.

Also, I don't agree with first reviewer's request for a GUI. Developing one is a significant task that requires a skill set distinct from the analysis done here. Also, early adopters, who will give a tool the scrutiny necessary to determine if it should be more widely adopted, really need to get "under the hood" and play with the code. As impressed as I am, the ultimate test is whether a community of early adopters likes it enough to play with it.

Suggestions:

- 1) Figure S1: Horizontal scale is NDD when it should be NND
- 2) Figure 2H: Could you put in a horizontal dashed line for 1 like you do in some of the supplemental figures?
- 3) Page 2, lines 44-45: The subject is the singular noun "value" but the verbs are "incorporate" and "compress." They should be "incorporates" and "compresses."

Reviewer #3:

Remarks to the Author:

Ok, yes, I take the authors point about novelty. I might have been too negative before. I would recommend the paper is accepted

The authors have addressed most of my other comments, however I would recommend a discussion on the multiple-blinking - basically a warning that if this artefact has not been corrected the colocalisation results will be skewed. The authors could also cite some literature on how this correction can be done. Annibale Plos ONE 2011 is probably the most cited but Bohrer Nature Methods 2021 and Jensen Biorxiv 2021 are the most up to date

Reviewer #4:

None

RESPONSE TO REVIEWERS

(NCOMMS-22-03147-A Ejdrup et al., Relative enrichment – a density-based colocalization measure for single-molecule localization microscopy).

Reviewer #1

We have happy that the reviewer feels we have answered their satisfactorily and agree that size is a matter of perspective.

The authors have answered my queries and criticisms satisfactorily. And indeed, whether size matters or not is a question of perspective... They should carefully checked the new references added as not all of them are correct.

We have now checked all references again and believe the three added in the previous revision are correct.

Reviewer #2

We greatly appreciate the reviewer noting the expansion of the manuscript and rereading with a fresh perspective. The review process has significantly improved the manuscript, and we are happy to be recommended for publication.

This manuscript has been somewhat expanded since the first submission. Between the new material and the fresh perspective of reading again later, I concede that this manuscript presents a significant methodological improvement over existing tools. The parameter-free nature, and robustness of results against swapping between reference molecule and molecule of interest, really is something significant. I particularly like that the authors are able to very clearly establish the expected quantitative trends for perfect co-localization, complete independence, and anti-correlation, and show it in multiple sets of real data. I recommend publication with only minor edits.

Also, I don't agree with first reviewer's request for a GUI. Developing one is a significant task that requires a skill set distinct from the analysis done here. Also, early adopters, who will give a tool the scrutiny necessary to determine if it should be more widely adopted, really need to get "under the hood" and play with the code. As impressed as I am, the ultimate test is whether a community of early adopters likes it enough to play with it.

Great point regarding the creation of a GUI. Early adopters should hopefully scrutinize the method for drawbacks and potential improvements.

Suggestions:

1) Figure S1: Horizontal scale is NDD when it should be NND

Good catch. We have corrected the mistake.

2) Figure 2H: Could you put in a horizontal dashed line for 1 like you do in some of the supplemental figures?

Excellent suggestion. We have added a horizontal line to the suggested figure (2H).

3) Page 2, lines 44-45: The subject is the singular noun "value" but the verbs are "incorporate" and "compress." They should be "incorporates" and "compresses."

Keen eye. We have corrected the grammatical error.

Reviewer #3

Once again, we are happy with the positive feedback after a round of revisions and the reviewer's openness to changing their minds. We thank the reviewer for the recommendation to publish.

Ok, yes, I take the authors point about novelty. I might have been too negative before. I would recommend the paper is accepted

The authors have addressed most of my other comments, however I would recommend a discussion on the multiple-blinking - basically a warning that if this artefact has not been corrected the colocalisation results will be skewed. The authors could also cite some literature on how this correction can be done. Annibale Plos ONE 2011 is probably the most cited but Bohrer Nature Methods 2021 and Jensen Biorxiv 2021 are the most up to date

We see the reviewer's point and have added a word of warning at the end of the discussion followed by references to suggested articles mentioned (line 339-340).